# Reversible Cerebral Vasoconstriction Syndrome Associated with Levonorgestrel-Releasing Intrauterine System

**DOI:** 10.3390/brainsci11050601

**Published:** 2021-05-08

**Authors:** Sangwon Choi, Ju-Young Lee, Jong Seok Bae, Hong-Ki Song, Ju-Hun Lee, Yerim Kim

**Affiliations:** 1Department of Neurology, Chuncheon National Hospital, Chuncheon 24409, Korea; chsangwon37@gmail.com; 2Department of Neurology, Kangdong Sacred Heart Hospital, Hallym University College of Medicine, Seoul 05355, Korea; wndud1030@gmail.com (J.-Y.L.); jsbae69@gmail.com (J.S.B.); hksong0@hanmail.net (H.-K.S.); leejuhun@kdh.or.kr (J.-H.L.)

**Keywords:** reversible cerebral vasoconstriction syndrome, endothelin-1, antioxidants, vasoconstriction, nitric oxide, vasodilation, brain, magnetic resonance angiography

## Abstract

Reversible cerebral vasoconstriction syndromes (RCVS) is a rare disease that is characterized by reversible multifocal stenosis of the cerebral arteries with various clinical manifestations. Though the pathomechanism of RCVS was unclear, we reported RCVS related to the levonorgestrel-releasing intrauterine system (IUS). A previous healthy 36-year-old woman had thunderclap headache after implanting the levonorgestrel-releasing IUS a year ago. In the serial angiography, we initially found left vertebra artery (VA), and then additionally new stenosis of both anterior cerebral arteries and middle cerebral arteries (MCA). Bilateral MCA stenosis improved but developed stenosis of right VA after a week. The mean flow velocities of both MCA increased in the first transcranial doppler (TCD), but normalized in the follow up TCD. Levonorgestrel might act as the vasoconstrictitve factor that increased the level of endothelin-1, diminished the release of NO and raised oxidative low-density lipoprotein (LDL). Although the exact pathological mechanisms for RCVS were not yet elucidated, this case might help clinicians understand the mechanisms of RCVS.

## 1. Introduction

Reversible cerebral vasoconstriction syndromes (RCVS) is a disease that shows reversible multifocal narrowing of the cerebral arteries with clinical manifestations [1]. The incidence rate is highest in the mid-40s and occurs more in women than in men [1]. The most common manifestation is severe headache, such as thunderclap headache, with or without neurologic symptoms [1,2]. Despite the neurological significance, the exact pathomechanism of RCVS remains enigmatic. It is also important to distinguish RCVS from other diseases including artery dissection or ruptured aneurysm that may manifest as similar clinical features and find out triggering factors such as vasoactive substances [2]. Furthermore, since local neural activity elicits cerebral blood flow variations, functional magnetic resonance imaging (fMRI) or electroencephalography might be one of several methods of data acquisition to investigate the cerebral hemodynamic response [3]. However, because vasoconstriction occurs dynamically in RCVS, such tests do not directly reflect hemodynamic changes and angiography or transcranial Doppler (TCD) findings are rarely supported. Here, we report a novel case of RCVS associated with the levonorgestrel-releasing intrauterine system (IUS) with evidence of imaging findings.

## 2. Case Presentation

A 36-year-old woman presented with a thunderclap headache every day from three days before hospitalization. She had not been diagnosed with migraine. She had had severe headaches once or twice a month since the levonorgestrel-releasing IUS was inserted in November 2016. When she had a headache, she took nonsteroidal anti-inflammatory drugs (NSAIDS) or acetaminophen. Computed tomography angiography (CTA) revealed segmental stenosis of the left vertebra artery (VA) (Figure 1A). In TCD, increased mean flow velocities (MFV) were recorded from both middle cerebral arteries (Figure 1D and Table 1). Two days later, magnetic resonance angiography (MRA) showed newly developed multiple arterial stenosis on both anterior cerebral arteries (ACA) of left A1 segment, right proximal A2 segment, and both middle cerebral arteries (MCA) (Figure 1B). In an additional TCD examination, the MFV of the right MCA and left ACA was increased in accordance with the MRA results (Figure 1E and Table 1). One week later, while follow up CTA also showed stenosis on the right ACA and a newly developed stenosis on the right VA, stenosis on both MCAs were resolved (Figure 1C). At follow up TCD after 1 month, the MFV of both MCA and left ACA was normalized in accordance with the CTA results (Figure 1F and Table 1). We diagnosed as reversible cerebral vasoconstriction syndrome and considered that RCVS is associated with the levonorgestrel-releasing IUS.

We prescribed nimodipine and the levonorgestrel-releasing IUS was removed from her. The headache gradually improved and follow-up TCD results also showed improvements (Table 1), and multiple arterial stenosis disappeared in one year follow-up MRA (Figure 2).

## 3. Discussion

Reversible cerebral vasoconstriction syndrome (RCVS) is a rare condition that occurs as the result of a sudden constriction of the vessels that supply blood to the brain [1]. The exact incidence rate is not clear, but the average age was about 40 years, and it has been reported to be more common in women than men [4,5]. The syndrome had various clinical features, which were acute onset and gradually resolved within several months [1,5]. The most common symptoms are thunderclap headaches, especially starting in the posterior head and occurring on both sides, sometimes with migraine-like symptoms [1,2]. Other neurologic symptoms are seizure and focal transient or lasting neurologic deficits including unilateral sensory symptoms, aphasia and hemiparesis [2].

Although the pathophysiological mechanisms of RCVS are not clearly elucidated, several possible mechanisms are suggested [2]. The one of hypothetical cause was representatively vasoactive agents leading to vasoconstriction which include illicit drugs, selective serotonin reuptake inhibitor (SSRI) and α-sympathomimetics [1,2,6]. There were additional factors to affect the vasoactivity; estrogen, prostaglandins, endothelin-1, serotonin, and nitric oxide [6,7]. RCVS has also been associated with disturbing factors of cerebral vascular tone such as sympathetic overactivity, endothelial dysfunction, and abnormal oxidative stress [6].

To our best knowledge, there has been no RCVS report related to the levonorgestrel-releasing IUS.

Levonorgestrel is a progestin or a synthetic form of the naturally occurring female sex hormone, progesterone [8]. Although the exact mechanism remains elusive, the mechanisms by which levonorgestrel may cause vasoconstriction are suggested as follows: (1) Endothelin-1 decreases during the menstrual cycle when estrogen is elevated [9]. Endothelin-1 is an endothelial-derived vasoconstricting substance [10]. Levonorgestrel reduces estrogen secretion, which increases the production of endothelin-1 and can cause vasoconstriction [9,10]. (2) Maintenance of nitric oxide (NO) bioavailability through the activation of endothelial NO synthesis is essential for maintaining vascular homeostasis [9,11]. Estrogen causes vasodilation by increasing the endothelial NO synthase expression and NO production [12]. Progestin inhibits estrogen and thus restrains NO function [9]. The deleterious effects of levonorgestrel on endothelial function may be related with its vasoconstricting properties caused by a decrease in NO release [13]. (3) Estrogen has an antioxidant effect [14]. Progestin inhibits estrogen, thereby increasing oxidative low-density lipoprotein [15]. This might lead to an increase in systemic oxidative stress, which can cause vasoconstriction [11,15].

## 4. Conclusions

To the best of our knowledge, this is the first report to investigate the dynamic vasoconstriction related with the levonorgestrel-releasing IUS. Levonorgestrel might contribute to vasoconstriction in terms of increasing or decreasing the release of endothelin-1, NO and oxidative low-density lipoproteins. This report is the first to suggest the possibility that levonorgestrel secreted through the IUD, rather than direct injection or oral administration, affect blood vessels, and is significant in that blood vessel changes were suggested through consecutive TCD and brain MRA. Although the exact etiology was not elucidated yet, this case might contribute to the understanding of the pathomechanism of RCVS.

## Figures and Tables

**Figure 1 brainsci-11-00601-f001:**
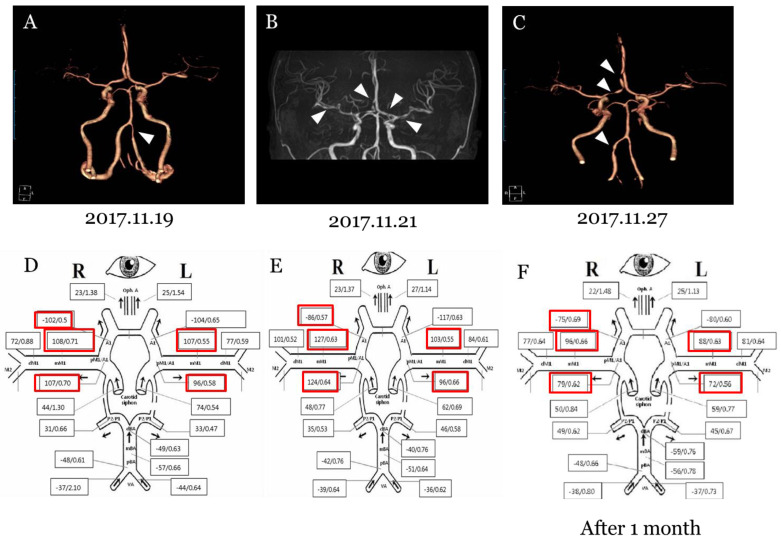
Serial images and TCD findings on patients. (**A**) Computed tomography angiography revealed segmental stenosis of the left vertebra artery (arrow). (**B**) Magnetic resonance angiography showed newly developed multiple arterial stenosis on left A1 segment, right proximal A2 segment of right anterior cerebral artery of, and bilateral middle cerebral arteries (arrows). (**C**) Follow up computed tomography angiography showed arterial stenosis on the right anterior cerebral artery and vertebral artery but the stenosis of both MCAs was improved. (**D**) Initial TCD showed increased MFV of bilateral MCAs and ACAs. (**E**) It was confirmed that the MFV of right MCA and left ACA was worse in the subsequent TCD. (**F**) The last TCD after 1 month showed that MFV of bilateral MCAs and ACAs was improved.

**Figure 2 brainsci-11-00601-f002:**
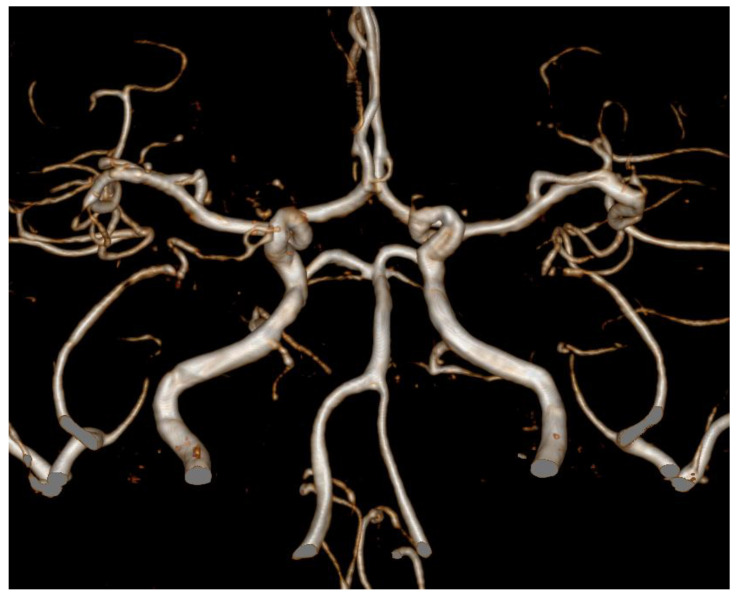
Magnetic resonance angiography (MRA). One year follow-up MRA revealed that multiple arterial stenosis was resolved.

**Table 1 brainsci-11-00601-t001:** Serial transcranial doppler (TCD) findings on patient.

	2017.11.20		2017.11.21		2018.01.16	
	MFV	PI	MFV	PI	MFV	PI
	Rt/Lt	Rt/Lt	Rt/Lt	Rt/Lt	Rt/Lt	Rt/Lt
ACA	102/104	0.55/0.65	86/117	0.57/0.63	75/80	0.69/0.60
MCA	114/107	0.70/0.55	127/103	0.63/0.55	94/88	0.65/0.63
PCA	31/33	0.66/0.47	35/46	0.53/0.58	49/45	0.62/0.67
VA	37/44	2.10/0.64	39/36	0.64/0.62	38/37	0.80/0.73
BA	57	0.66	51	0.64	59	0.76

Abbreviation: Mean Flow Velocities: MFV, Pulsatile index: PI, Rt: Right, Lt: Left, Anterior Cerebral Artery: ACA, Middle Cerebral Artery: MCA, Posterior Cerebral Artery: PCA, Vertebral Artery: VA, Basilar Artery: BA.

## Data Availability

All data generated or analyzed during this study are included in this published article. Anonymized data will be shared by reasonable request from any qualified investigator.

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
