# Peer review of "Reversible Cerebral Vasoconstriction Syndrome Associated with Levonorgestrel-Releasing Intrauterine System"

_brainsci, 2021, doi:10.3390/brainsci11050601_

Round 1

Reviewer 1 Report

There is no doubt about reversible cerebral vasoconstriction syndromes is a rare disease that characterized reversible multifocal stenosis of the cerebral arteries with various clinical manifestation. Authors in article reported RCVS related to the levonorgestrel-releasing intrauterine system. 

My comments to the article are as follows:

- The document lacks a clear record of when exactly the case took place.

- On lines 39 and 48 there is no reference to the table number

- The introduction to the article is sparse, despite the fact that it is a Case Report. I propose to extend the Introduction by connecting references to one of the publications on the functioning / acquisition of brain data, for example: Data Acquisition Methods for Human Brain Activity,, Analysis and classification of eeg signals for brain-computer interfaces, Book Series: Studies in Computational Intelligence, Springer from 2020.

- Please describe in Conclusion how you think this case will contribute to the understanding of the RCVS.

Reviewer 2 Report

The auhors present a case of reversible cerebral vasoconstriction syndromes (RCVS), which presented with thunderclap headache, in a 36-year-old woman. They postulated that the RCVS was secondary to a levonorgestrel-releasing IUS implanded one year before.  

The case is intriguing because RCVS is probably underestimated but potentially relevant cause of stroke; it is characterized by a usually (but not always) reversible multifocal stenosis of the intracranial arteries, and its exact pathogenetic mechanism is unclear.

The authors postulated that the Levonorgestrel could act as a vasoconstrictitve factor, which increases the level of endothelin-1 and reduces the release of NO and raised oxidative low-20 density lipoprotein (LDL). 

Despite the casesis well described, I have several relevant issues about the manuscript.

The first one is related to the clinical hystory of the patients, affected by migraine ( "severe headaches once or twice a month"), so not properly "previously health".. It should be ineresting to know if she used to take, i.e., triptans; these drugs could act as vascular modulators. Moreover, migraine per se could be related to RCVS. Did he authors also excluded the possible use of cannabis or others illicit drugs?

It is also uclear if sporadic segmental stenosis were present in previous neuroradiological imaging, if ones (but migrain patients usually are submitted to at least one MRI or CT scan, during their life). This point is not strictly relevant, but it could be interesting to understand it..

An interesting, recent point of view has been presented by Cho S et al (RCVS-TCH score can predict reversible cerebral vasoconstriction syndrome in patients with thunderclap headache. Sci Rep. 2021 Apr 8;11(1):7750. doi: 10.1038/s41598-021-87412-7) about the RCVS. 

Does nimodipine or other drugs have been proposed to the patients? In order to reduce the segmental stenosis

Hovewer, my main criticism reading the manuscript is related to the pathogenetic putative mechanism...despite interesting, the proposed mechanism is speculative...

Round 2

Reviewer 2 Report

The manuscript has been modified...and I think that now it could be more interesting for readers

This manuscript is a resubmission of an earlier submission. The following is a list of the peer review reports and author responses from that submission.